

# Proton Plasma Asymmetries between the Convective-Electric-Field Hemispheres of Venus' Dayside Magnetosheath

Sebastián Rojas Mata[1], Gabriella Stenberg Wieser[1], Tielong Zhang[2], and Yoshifumi Futaana[1]

[1]Swedish Institute of Space Physics, Kiruna, Sweden
[2]Space Research Institute, Austrian Academy of Science, Graz, Austria

**Correspondence:** Sebastián Rojas Mata (sebastian.rojasmata@irf.se)

**Abstract.** Proton plasma asymmetries with respect to the convective electric field ($\mathbf{E}$) are characterized in Venus' dayside magnetosheath using measurements taken by an ion mass-energy spectrometer and a magnetometer. Investigating the spatial structure of the magnetosheath plasma in this manner provides insight into the coupling between solar-wind protons and planetary ions. A previously developed methodology for statistically quantifying asymmetries is further developed and applied to an existing database of proton bulk-parameter measurements in the dayside magnetosheath. The density and speed exhibit weak asymmetries favoring the hemisphere in which $\mathbf{E}$ points towards the planet, while the magnetic-field strength has a weak asymmetry favoring the opposite hemisphere. The temperatures perpendicular and parallel to the background magnetic field as well as their ratio present no significant asymmetries. Deflection of the solar wind due to momentum exchange with planetary ions is revealed by (1) the $O^+$ Larmor-radius trends of the asymmetries of the bulk-velocity components perpendicular to the upstream solar-wind flow and (2) the $\mathbf{E} \times \mathbf{B}_{IMF}$-drift trends of the bulk-velocity component along the cross-flow component of the interplanetary magnetic field ($\mathbf{B}_{IMF}$). These interpretations are enabled by comparisons to studies of solar-wind deflection at Mars and comet 67P/Churyumov-Gerasimenko, highlighting the benefits of comparative planetology studies.

## 1  Introduction

Unmagnetized bodies like Venus and Mars experience a closer interaction with the solar wind than those with an intrinsic magnetic field (Russell et al., 2016; Futaana et al., 2017). The upstream interplanetary magnetic field (IMF) $\mathbf{B}_{IMF}$ and convective electric field $\mathbf{E} = -\mathbf{v}_{SW} \times \mathbf{B}_{IMF}$ (where $\mathbf{v}_{SW}$ is the solar-wind velocity) influence several phenomena such as plasma boundary morphology (Phillips et al., 1986; Zhang et al., 1991b; Edberg et al., 2009; Chai et al., 2015; Signoles et al., 2023), pick-up ion dynamics (Barabash et al., 2007b, a; Brain et al., 2016; Jarvinen et al., 2016), and plasma wave activity (Du et al., 2010; Delva et al., 2011; Ruhunusiri et al., 2017; Xiao et al., 2018). In particular, the structure of the magnetosheath, the region where the solar wind transfers momentum and energy to the planet's magnetosphere (Longmore et al., 2005; Lucek et al., 2005; Haaland et al., 2017), is responsive to the configuration of the upstream electromagnetic fields. For example, the orientation of $\mathbf{B}_{IMF}$ with respect to the bow shock normal affects solar-wind proton flows and temperature anisotropies (Halekas et al., 2017; Rojas Mata et al., 2023). More generally, observational studies of the magnetosheaths properties at bodies across the Solar System reveal significant dependencies on (magnetic) longitude, commonly referred to as dawn-dusk or $q_\perp/q_\parallel$ asymmetries



(Dubinin et al., 2008; Dimmock and Nykyri, 2013; Walsh et al., 2014; Haaland et al., 2017; Carbary et al., 2017; Palmaerts et al., 2017; Behar et al., 2018; Rojas Mata et al., 2023). Investigating the physics behind these asymmetries has meaningfully advanced our fundamental understanding of the solar-wind interaction with the different bodies.

In contrast, few studies investigate analogous solar-wind plasma asymmetries at unmagnetized bodies as a function of latitude, i.e. between the hemispheres in which the convective electric field points away from ($+E$) or towards ($-E$) the 30 body. At Mars, Dubinin et al. (2018) found that the magnetosheath plasma in the $+E$ hemisphere is slower and more deflected in the direction opposite to $\mathbf{E}$. Romanelli et al. (2020) determined that this asymmetry decreases with respect to the solar-wind density and increases with respect to the cross-flow component of $\mathbf{B}_{IMF}$, which is consistent with a two-fluid description of mass loading by pick-up ions. Similar analyses based on plasma data at Venus have not been found, while magnetometer-based investigations found that the magnitude of $\mathbf{B}_{IMF}$ throughout the magnetic barrier is greater in the $+E$ hemisphere (Phillips 35 et al., 1986; Zhang et al., 1991b; Du et al., 2013; Xiao et al., 2018). Additionally, at both planets $\mathbf{B}_{IMF}$ wraps asymmetrically (e.g. more tightly in the $-E$ hemisphere) in the magnetosheath and magnetotail (Zhang et al., 2010; Du et al., 2013; Dubinin et al., 2019, 2021; Zhang et al., 2022). Other studies which mention differences between the $\pm E$ hemispheres instead focus on the dynamics of pick-up ion escape (Barabash et al., 2007b, a; Dubinin et al., 2011; Xu et al., 2023), which at Venus has been linked to the $\mathbf{B}_{IMF}$ asymmetries (Luhmann et al., 1985; Phillips et al., 1987).

In parallel, numerical studies of the plasma environment around unmagnetized bodies have investigated topics such as hemispheric asymmetries, plasma boundary morphology, and pick-up ion dynamics (Brecht and Ferrante, 1991; Moore et al., 1991; Shimazu, 1999; Kallio et al., 2011). For example, recent hybrid models (Kallio et al., 2006; Jarvinen et al., 2013, 2016) reproduce observed $\pm E$ asymmetries concerning plasma velocities and magnetic fields, as well as indicate that the dynamics of pick-up ions depend on their upstream Larmor radius

$$45 \quad r_{L,i} = \frac{m_i}{q_i} \frac{|\mathbf{v}_{SW}| B_y}{B_{IMF}^2}, \tag{1}$$

where $m_i$ is the ion mass, $q_i$ the ion charge, and $B_y$ the cross-flow component of $\mathbf{B}_{IMF}$. Though the asymmetries and pick-up ions may be linked, their exact interdependence remains unresolved as asymmetries arise even if planetary ions ($O^+$ and $H^+$) are not included in the simulation (Brecht, 1990; Jarvinen et al., 2013). Additionally, few studies directly compare global simulations to local spacecraft data, so how well the models quantitatively reproduce the measured spatial structure of the 50 plasma environment is not fully determined.

The above illustrates the opportunity to develop new insight into magnetosheath physics by comparing observations and simulations not only at a single body, but also across different bodies (i.e. Venus, Mars, and even comets). To this end, in this paper we characterize the proton plasma asymmetries between the $+E$ and $-E$ hemispheres of Venus' dayside magnetosheath. We apply and extend the methodology Rojas Mata et al. (2023) developed to statistically quantify asymmetries. In Sect. 2 we 55 overview the data set used as well as the methodology for quantifying parameter asymmetries; our results follow in Sect. 3. We discuss connections to relevant numerical and observational studies in Sect. 4 and present concluding remarks in Sect. 5.





## 2 Data and methodology

### 2.1 Dayside magnetosheath database

For this study we use a database of proton plasma bulk-parameter measurements in Venus' dayside magnetosheath (Rojas Mata
and Futaana, 2022). Based on measurements taken by the Ion Mass Analyser (IMA) instrument (Barabash et al., 2007c) and
the Magnetometer (MAG) (Zhang et al., 2006) on board the Venus Express (VEX) mission (Svedhem et al., 2007), the database
includes densities, velocities, and both perpendicular and parallel temperatures for 1181 locations in the magnetosheath along
with the upstream solar wind conditions for the 597 orbits spanned. These bulk parameters result from bi-Maxwellian gy-
rotropic fits to IMA's velocity-distribution-function measurements (Bader et al., 2019; Rojas Mata et al., 2022). As detailed
in Rojas Mata et al. (2023), the database includes data only from orbits with identifiable dayside bow-shock crossings and
well-defined upstream solar wind conditions.

To characterize asymmetries between the $\pm E$ hemispheres of the magnetosheath we use the Venus-Sun-Electric field (VSE)
reference frame. The $+X_{VSE}$ axis points against the solar-wind velocity, whose aberration we correct for using each orbit's
upstream measurements (Rojas Mata et al., 2023). The $+Y_{VSE}$ axis points along the cross-flow component of $\mathbf{B}_{IMF}$, making
$+Z_{VSE}$ point along the upstream convective electric field $\mathbf{E} = -\mathbf{v}_{SW} \times \mathbf{B}_{IMF}$. The $\pm E$ hemispheres then correspond to the
north ($+Z_{VSE}$) and south ($-Z_{VSE}$) hemispheres of this reference frame. The spatial coverage of the measurements in this
reference frame is decently uniform except for limited coverage on the dayside close to the subsolar point caused by VEX's
orbit geometry (see the 'subsolar-wind hole' in Fig. 4 of Rojas Mata et al. (2023)).

### 2.2 Statistically quantifying asymmetries

Since VEX's highly-elliptical, quasi-polar orbit lead to the sampling of opposing VSE hemispheres under different solar-wind
conditions, measurement-by-measurement pairing is not possible in order to quantify spatial asymmetries. Rojas Mata et al.
(2023) addressed this challenge by developing a methodology which uses distributions of ratios of estimated measurement
distributions as measures of the plasma parameter asymmetries. The technique also quantifies the variability of the asymme-
tries, provides flexibility for analyzing spatially binned data, and does not rely on models for the distributions. We refer the
reader to the reference for the full discussion of the methodology; here we briefly overview the steps for quantifying the spatial
asymmetry of a parameter $a$ (e.g. speed or magnetic-field strength) between hemispheres $H1$ and $H2$:

1. Normalize the measurements of $a$ by their corresponding value in the upstream solar wind, i.e. $\widehat{a} = a/a_{SW}$

2. Approximate the probability distribution function (PDF) of $\widehat{a}$ in each hemisphere using Gaussian kernel density estimates

3. Draw $\mathcal{O}(10^6)$ random samples each of $\widehat{a}_{H1}$ and $\widehat{a}_{H2}$ using the estimated PDFs

4. Compute the distribution of $\overline{a} = \widehat{a}_{H1}/\widehat{a}_{H2}$ by pairing the samples

In this work $H1 = +E$ hemisphere and $H2 = -E$ hemisphere of the dayside magnetosheath. The process can also be applied
to binned data (e.g. between two bins centered at corresponding latitudes in each hemisphere), so we also compute asymmetries



with the measurements sorted in 15°-wide latitudinal bins with a 50% overlap. Note that such binning means that we average over radial distance and longitude.

## 3 Results

### 3.1 Scalar parameters

We present in Fig. 1 the medians of the magnetosheath measurements (top row) and of their normalized values (bottom row) as of function of VSE latitude. The 'error' bars indicate the first and third quartiles of the measurement distributions in each bin; rather than uncertainty, these values reflect the spread of the distributions and how they shift across bins. Bins centered less than 60° (45°) from the central parallel contain more than 70 (100) scans each. Bins 75° or farther contain much fewer scans ($< 25$) so these data may have lower statistical reliability.

The plasma speed appears lower closer to the central parallel, which is consistent with the expectation of higher deceleration closer to the near-sub-solar-wind region (Spreiter and Alksne, 1966; Spreiter et al., 1970). Other parameters do not seem to exhibit clear trends as a function of latitude, especially given their wide variability. We display in Fig. 2 the median parameter asymmetries as a function of latitudinal distance from the central parallel. Additionally, the top marker in each plot indicates the overall asymmetry calculated using data across all latitudes in each hemisphere. Despite the larger asymmetries observed in the bins above 60° (which contain much fewer measurements than the others), most parameters exhibit weak or insignificant asymmetries. The plasma speed is higher in the $-E$ hemisphere while the IMF magnitude is slightly higher in the $+E$ hemisphere. Both of these asymmetries have been observed before (Phillips et al., 1986; Zhang et al., 1991b; Du et al., 2013; Dubinin et al., 2018; Xiao et al., 2018; Xu et al., 2023). The slightly lower density observed in the $+E$ hemisphere maybe has connections to studies about plasma depletion in the magnetic barrier (Zwan and Wolf, 1976; Zhang et al., 1991a; Luhmann, 1995); even if we did not average over radial distance, the spatial resolution of the IMA scans (0.2-0.3 $R_V$ with $R_V$ the Venus radius) is insufficient to properly discern this effect given the expected thickness of such a plasma depletion layer (less than 1000 km). The perpendicular and parallel temperatures exhibit the clearest symmetry between hemispheres (both overall and as a function of latitude). The temperature anisotropy exhibits more variability but overall there still does not seem to be any significant asymmetry.

These results contrast with the asymmetries between magnetosheaths downstream of different bow-shock geometries (see Fig. 6 in Rojas Mata et al. (2023)) which are more significant and exhibit trends with upstream Alfvén Mach number. This indicates that the convective electric field has little influence on average magnetosheath properties, especially compared to the bow shock geometry. Given the potential connection between $\pm E$ asymmetries and pick-up ion dynamics, we also checked for dependencies on upstream $O^+$ Larmor radius yet none of the parameters exhibited a significant trend. However, since varying the upstream Larmor radius changes the direction of the pick-up ion trajectories (Jarvinen et al., 2016), analyzing the components of the bulk velocity likely yields more informative results on the matter. This requires us to to reevaluate our methodology to make it adequate for quantities which are not strictly positive.





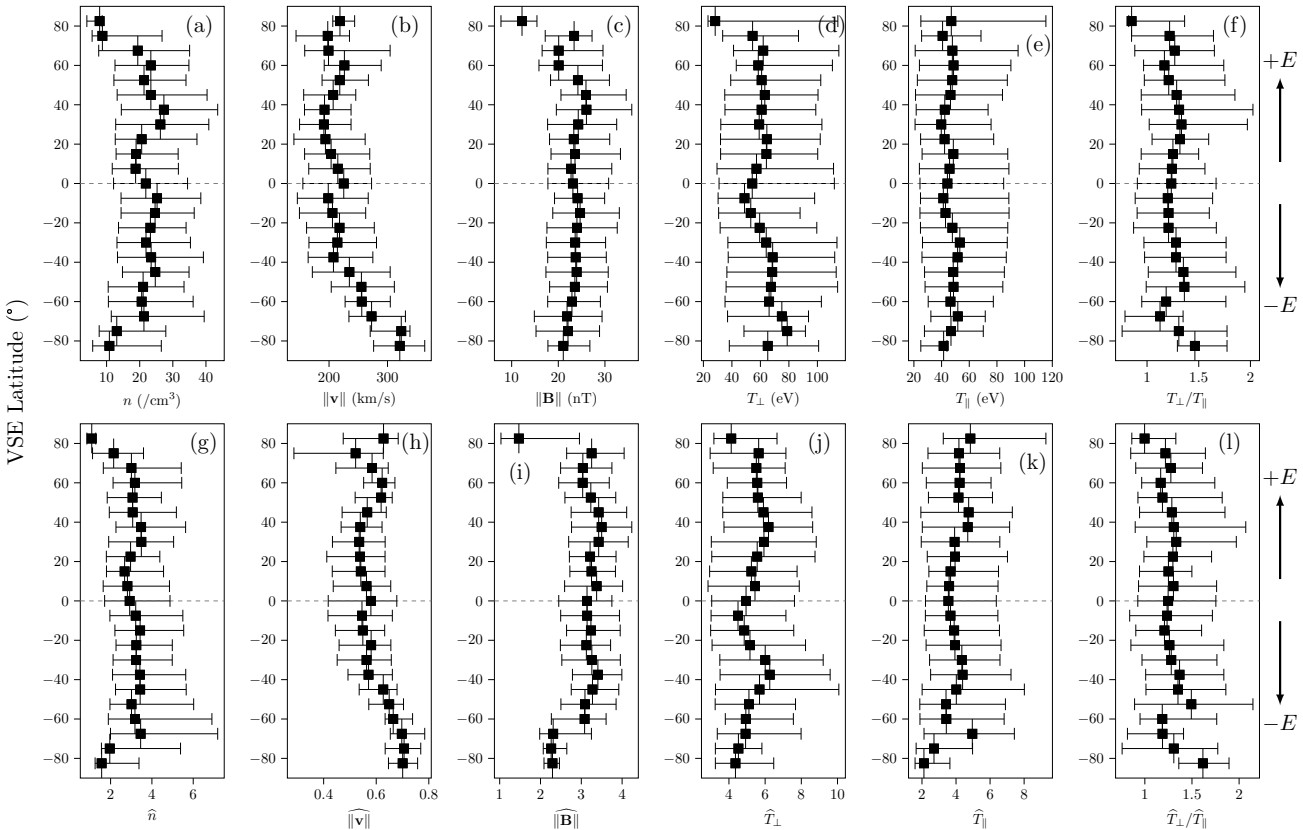

**Figure 1.** Proton parameters as a function of latitudinal distance from the central parallel. The top row shows unnormalized values, the bottom row normalized by the solar-wind value. Positive VSE latitude corresponds to the $+E$ hemisphere, negative to the $-E$ hemisphere. Markers indicate medians while 'error' bars correspond to the first and third quartiles.

## 3.2 Bulk-velocity components

We here consider the measurement distributions of $v_x$, $v_y$, and $v_z$ in the $+E$ and $-E$ hemispheres presented in Fig. 3. We also display the distributions for the data subsets corresponding to large (above the third quartile 1.58 $R_V$) and small (below the first quartile 0.75 $R_V$) upstream $O^+$ Larmor radius $r_{L,O^+}$ (the median for all data is 1.09 $R_V$). We also indicate the median and quartiles for each distribution at the top of each plot. We note that only $v_x$ could be normalized by its upstream value since the solar wind points solely along $X_{VSE}$; we thus only consider unnormalized measurements. As expected, $v_x$ is negative and larger than the other components. $v_y$ is unevenly distributed about 0 due to a previously observed asymmetry in the proton flow in the VSO frame attributed to the planet's orbital motion (Lundin, 2011). Since VEX only sampled the northern VSO hemisphere, this asymmetry does not average out when converting into the VSE frame, leading to mostly positive (negative) values in the $+E$ ($-E$) hemisphere. Finally, while the sign of $v_z$ is mostly as expected for each hemisphere, IMA's limited



**Figure 2.** Proton parameter asymmetries as a function of latitudinal distance from the central parallel. The marker at the top corresponds to the overall asymmetry across all latitudes for each hemisphere. The asymmetry favors the $+E$ hemisphere if $\overline{a} > 1$ and the $-E$ hemisphere if $\overline{a} < 1$. Markers indicate medians while 'error' bars correspond to the first and third quartiles. Note the varying horizontal scales.





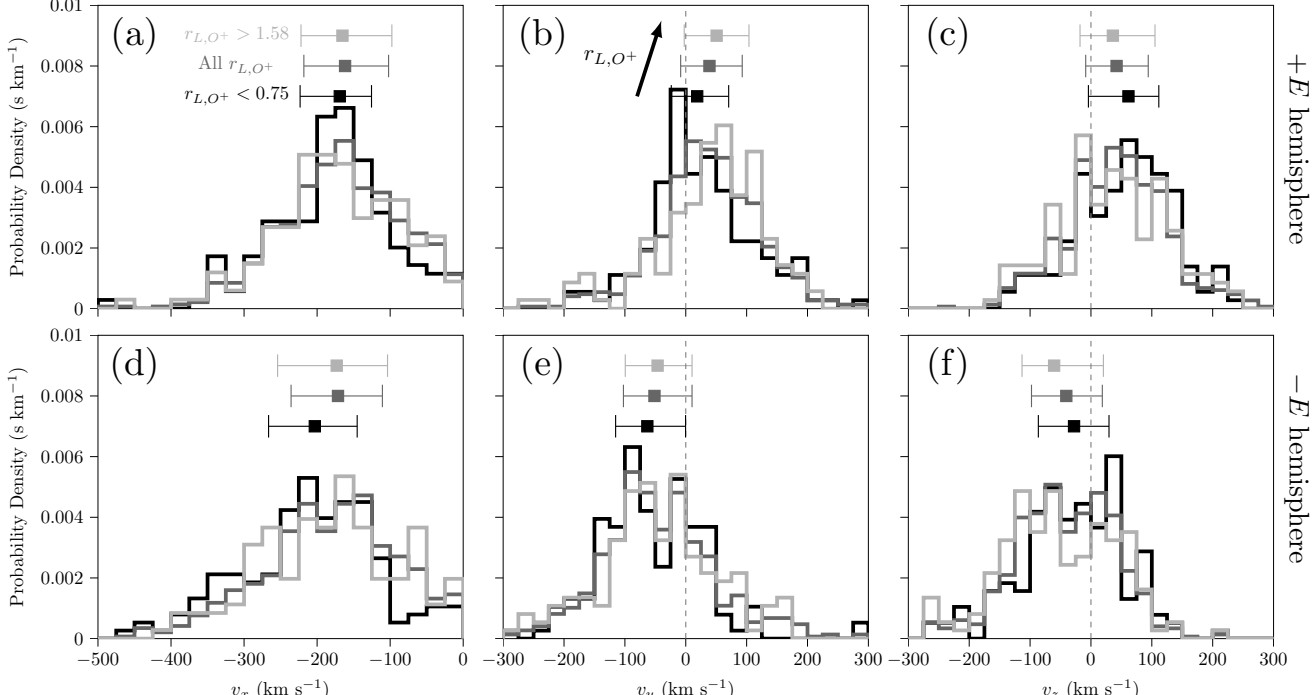

**Figure 3.** Distributions of the proton bulk velocity components in the $+E$ (top row) and $-E$ (bottom row) hemispheres. Black is for data with $r_{L,O+} < 0.75 R_V$, dark gray for all $r_{L,O+}$, and light gray for $r_{L,O+} > 1.58 R_V$. The markers indicate the median for each distribution along with the respective first and third quartiles as the 'error' bars. The arrow indicates increasing $r_{L,O+}$.

FOV combined with the spacecraft's orientation sometimes leads to measurements with poor constraints on $v_z$. Reviewing the measurements to correct $v_z$ is beyond the scope of this work; our methodology and the equal impact of systematic errors on both hemispheres mitigate potential errors anyway.

$v_x$ does not vary significantly with $r_{L,O+}$ in the $+E$ hemisphere. In the $-E$ hemisphere the small $r_{L,O+}$ measurements have a sudden dip around -100 km s$^{-1}$ (possibly a random sampling artifact) which, if not present, would also make $v_x$

insensitive to $r_{L,O+}$. Both $v_y$ and $v_z$ follow opposite trends with $r_{L,O+}$ regardless of hemisphere: $v_y$ becomes more positive as $r_{L,O+}$ increases and $v_z$ more negative. The solar wind thus seems to deflect towards the $+Y_{VSE}$ and $-Z_{VSE}$ directions as $r_{L,O+}$ increases. Since pick-up ions are more common in the $+E$ hemisphere (Phillips et al., 1987; Barabash et al., 2007b; Jarvinen et al., 2013), quantifying asymmetries of $v_y$ and $v_z$ as a function of $r_{L,O+}$ may clarify how these trends relate to the momentum exchange between solar-wind protons and planetary ions. Previously, ratios provided easily interpretable measures

of asymmetry for positive scalar parameters such as density or temperature. However, the measurement distributions of $v_y$ and $v_z$ have positive and negative portions, so, while the methodology from Sect. 2.2 can still be applied, the resulting distributions of parameters ratios are more difficult to interpret. An alternative is to use a sum instead of a ratio as the measure of asymmetry; Romanelli et al. (2020) did so to study $\pm E$ asymmetries in $v_z$ at Mars. As the average $v_z$ was primarily positive in the $+E$





hemisphere and negative in the $-E$ one, summing the values between the hemispheres always gave the difference in the
magnitude of this component. We therefore modify the procedure from Sect. 2.2 to use sums of unnormalized parameters as
our measure of asymmetry, i.e. $\overline{a} = a_{H1} + a_{H2}$ instead of $\overline{a} = \widehat{a}_{H1}/\widehat{a}_{H2}$. Now $\overline{a} > 0$ indicates a $+E$-favored asymmetry and
$\overline{a} < 0$ a $-E$-favored one. For our data, the 'bodies' of the $v_y$ and $v_z$ distributions have opposite signs in opposite hemispheres,
so the 'bodies' of the sum distributions provides reliable measures of average asymmetries. However, as this is not true for the
'tails' of the distributions, the sum distributions may be artificially wide.

Adopting these modifications we present in Fig. 4 the medians of the $v_y$ and $v_z$ asymmetries across all latitudes in each
hemisphere. Again the 'error' bars indicate the first and third quartiles of the distributions. We also show the results for the data
subsets corresponding to large and small upstream $O^+$ Larmor radius. The $v_y$ asymmetry decreases with increasing $O^+$ Larmor
radius, suggesting that the underlying mechanism deflecting the solar wind in the $y$-direction either disappears or affects both
hemispheres more evenly as $r_{L,O^+}$ increases. Meanwhile, the $v_z$ asymmetry favors the $+E$ hemisphere for small $r_{L,O^+}$ and
the $-E$ hemisphere for large $r_{L,O^+}$. This switch in which hemisphere the asymmetry favors may be connected to how much
$\mathbf{E} \times \mathbf{B}$-drift and finite-Larmor-radius dynamics affect momentum transfer between planetary and solar-wind ions for different
$r_{L,O^+}$ (see Sect. 4). We note that these trends do not arise when splitting the data by high and low values for $|\mathbf{v}|$, $|\mathbf{B}_{IMF}|$, or
$B_y$, the individual parameters which comprise $r_{L,O^+}$.

## 4 Discussion

Our interpretation of these Larmor-radius-dependent trends of solar-wind deflection at Venus benefits from comparisons with
observations at Mars and comets; such comparative studies place the discussion into a broader context of solar-wind interactions
with unmagnetized atmospheric bodies (see, for example, Luhmann et al. (1987); Fedorov et al. (2008); Holmstrom and Wang
(2015); Jarvinen et al. (2016)). However, as mentioned before, few studies provide quantitative characterizations of plasma
asymmetries between the $\pm E$ hemispheres, let alone investigate dependencies on Larmor radius. This is understandable as the
Larmor radii of pick-up ions at these bodies are commonly larger than the obstacle radius. Thus, the particle motion is studied
in the large-Larmor-radius limit in which other parameters are relevant. This contrasts with the range of $r_{L,O^+}$ we observe at
Venus (about 0.4–2.4 $R_V$), which means the data likely covers mixed dynamical regimes. This is illustrated by simulations in
which pick-up ion species with Larmor radii similar to the planet radius experience both $\mathbf{E} \times \mathbf{B}$-drift and finite-Larmor-radius
dynamics (Jarvinen et al., 2016). Fundamental differences like these complicate but do not impede drawing beneficial insight
from comparisons between these bodies.

### 4.1 Comparison to Mars

We first compare our results to Romanelli et al. (2020)'s analysis of $\pm E$ asymmetries in the $z$-component of the proton bulk
velocity in Mars' magnetosheath. This component was greater in magnitude in the $-E$ hemisphere, coinciding with what
we observe for large $r_{L,O^+}$ at Venus. Using a two-species ion fluid description, the authors derived an analytic expression
"suggesting a dependence between the SW flow asymmetry on the $(eB_y)/(n_{SW}m_p)$ external factor", where $m_p$ is the proton





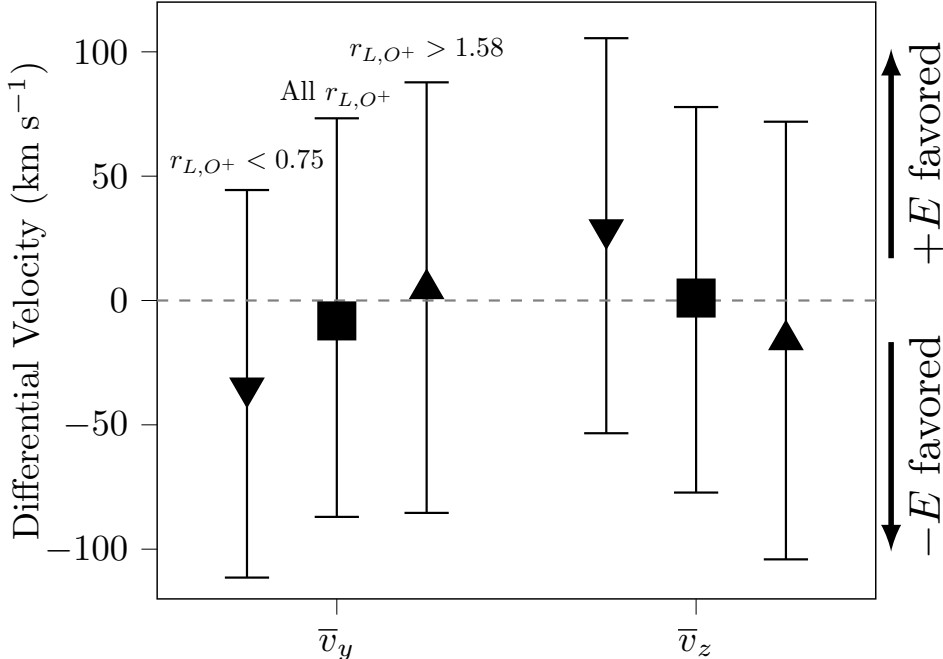

**Figure 4.** Proton bulk velocity component asymmetries as a function of upstream $O^+$ Larmor radius. For each parameter, the left marker is for $r_{L,O^+} < 0.75$, the middle for all $r_{L,O^+}$, and the right for $r_{L,O^+} > 1.58$. The asymmetry favors the $+E$ hemisphere if $\overline{a} > 0$ and the $-E$ hemisphere if $\overline{a} < 0$. Markers indicate the median for each distribution along with the respective first and third quartiles as the 'error' bars.

mass and $n_{SW}$ the solar-wind density. The measured distributions of $\overline{v}_z$ indeed confirm this predicted dependence of the asymmetry on $B_y$, $n_{SW}$, and $(eB_y)/(n_{SW}m_p)$. We do not observe similar trends with respect to these parameters in the $v_z$ asymmetry at Venus, simply verifying that the assumptions based on a large Larmor radius for planetary ions at Mars are not applicable. The authors did not investigate trends with upstream Larmor radius, thereby impeding further comparison with our analysis. Unfortunately no other studies have characterized proton plasma magnetosheath asymmetries at Mars; given the abundance of plasma data provided by missions like Mars Express (Chicarro et al., 2004; Barabash et al., 2006) or MAVEN (Jakosky et al., 2015; Halekas et al., 2015), future work could provide new beneficial insight into the $v_z$ asymmetry (or that of any parameter) by applying our methodology at Mars.

## 4.2 Comparison to simulations

As with observational work, existing numerical studies have not characterized $\pm E$ asymmetries in the magnetosheath plasma as a function of upstream Larmor radius. Nevertheless, Jarvinen et al. (2016)'s global hybrid simulations of planetary ion dynamics at Venus and Mars provide pertinent results to contextualize our observations. The authors simulated the planets' plasma environment under their respective nominal upstream conditions ("Venus nominal" and "Mars nominal"), but also with





the heliodistance of each planet interchanged ("Mars at Venus" and "Venus at Mars"). By analyzing test particle trajectories
of planetary ions with $m/q = 1$, 4, 16, and 32 (i.e. H$^+$, He$^+$, O$^+$, and O$_2^+$) released at an altitude of 0.2 planet radii (see
Fig. 6-9 in the reference), the authors investigated different factors affecting the $\mathbf{E} \times \mathbf{B}$-drift and finite-Larmor-radius dynamics
of escaping ions. All runs feature stronger magnetic fields and O$^+$ ions concentrated in the $+E$ hemisphere.

   The key difference in the four cases simulated is the increasing upstream O$^+$ Larmor radius of 0.7, 1.2, 3.0, and 5.3 planet
radii (see Table 2 in the reference), the parameter chosen as a "first approximation of how important the finite-Larmor-radius
effects are for the dynamics of escaping planetary ions" (Jarvinen et al., 2016). As $r_{L,O^+}$ increases in the simulations, the
O$^+$ trajectories in the $+E$ hemisphere align more along the $+Z_{VSE}$ axis; the pick-up ions accelerate less in the $-Y_{VSE}$
direction and more in the $+Z_{VSE}$ direction. With less pick-up ion motion along the $Y_{VSE}$ axis, differences in the solar-wind
$v_y$ distributions between the $\pm E$ hemispheres should decrease, which is precisely the trend we see in the data in Fig. 4a.
Simultaneously, the solar-wind $v_z$ distributions in both hemispheres should become more negative, which we see in Fig. 3cf.
Since $v_z$ is mostly positive in the $+E$ hemisphere and negative in the $-E$ one, the $v_z$ asymmetry becomes more $-E$ favored
as $r_{L,O^+}$ increases, as shown in Fig. 4a. So, despite our Venus data not spanning the same range of $r_{L,O^+}$ as the simulations, it
seems the Larmor-radius-dependent trends in the $\pm E$ asymmetries are consistent with varying momentum exchange between
planetary O$^+$ and solar-wind protons.

   Still unexplained, though, is why the $v_z$ asymmetry is $+E$ favored for small $r_{L,O^+}$. We have so far only considered momen-
tum exchange with heavy pick-up ions, yet the simulations show that the dynamics of lighter ions (like H+) are significantly
different since they experience more $\mathbf{E} \times \mathbf{B}$-drift dynamics than finite-Larmor-radius effects. Light ions concentrate in the $-E$
hemisphere, so their effect would be to deflect the solar wind towards the $+Z_{VSE}$ axis. Due to their smaller mass, this may
only be noticeable when the heavier ions also experience more $\mathbf{E} \times \mathbf{B}$-drift dynamics so that their contribution to momentum
exchange in the $Z_{VSE}$ direction is reduced or more even between hemispheres. New simulations expanding upon the results
in Jarvinen et al. (2016) and simultaneously quantifying plasma asymmetries could provide clarity to this matter.

## 4.3  Comparison to comet 67P/Churyumov-Gerasimenko

Behar et al. (2018)'s analysis of the solar-wind deflection at comet 67P/Churyumov-Gerasimenko provides guidance for further
interpreting the $v_y$ asymmetry. This statistical analysis of ion velocities measured during the entire Rosetta mission (Glassmeier
et al., 2007; Nilsson et al., 2007) shows that the solar-wind protons are deflected towards the dusk side of the comet. The authors
ascribe this effect to momentum exchange with cometary ions which instead drift towards the dawn side of the comet. These
results are valid for both inward and outward Parker spirals, indicating that fundamentally the cometary (solar-wind) ions are
deflected towards (away from) the $\mathbf{E} \times \mathbf{B}$ direction. Since the cross-flow component of the $\mathbf{E} \times \mathbf{B}$ direction is the same for
either inward or outward Parker spiral IMF, this leads to the consistent dawn/dusk deflections.

   Behar et al. (2018) do not separate the data by $\pm E$ hemispheres or Larmor radius, so a direct comparison with our analysis
so far is not possible. Their equations of generalized gyromotion indicate that the solar-wind deflection angle depends on the
ratios of mass and density between the solar-wind and cometary ions. While we have measurements of the solar-wind proton
density, we cannot readily implement reliable O$^+$ densities into our data set (which would serve as the analogous cometary





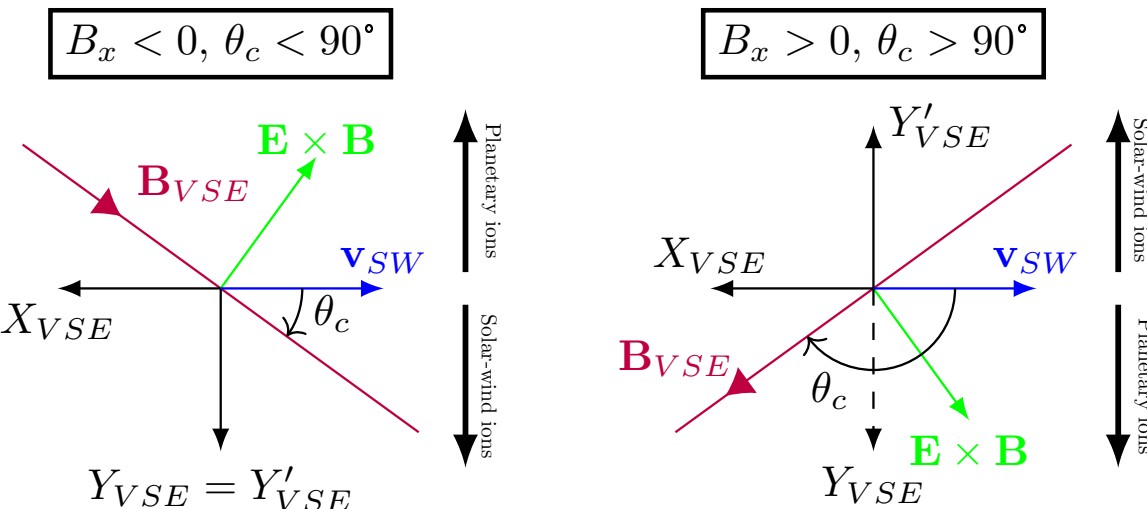

**Figure 5.** Orientation of the $\mathbf{E} \times \mathbf{B}$ direction and $Y'_{VSE}$ axis in 'Parker' (left) and 'ortho-Parker' (right) spiral configurations in the VSE frame. The black arrows indicate the directions in the $X_{VSE}$-$Y_{VSE}$ plane along which planetary ions drift and solar-wind ions deflect.

ion densities). However, we can conduct an exploratory analysis to determine if we detect similar solar-wind deflections with respect to $\mathbf{E} \times \mathbf{B}$ direction. We must first modify the VSE frame to define a $Y'_{VSE}$ axis which points anti-parallel to the

225 cross-flow component of the $\mathbf{E} \times \mathbf{B}$ direction. As illustrated in Fig. 5, inward Parker spirals in VSO (i.e. $B_x > 0$) become 'ortho-Parker' in VSE, so the solar-wind proton's $v_y$ should become more negative (as opposed to more positive as in Behar et al. (2018)). Thus, to facilitate comparisons with observations at comet 67P, we flip the sign of the $y$-component of velocity for measurements with $B_x > 0$. This means that momentum exchange with planetary ions moving in the $\mathbf{E} \times \mathbf{B}$ direction would always make the solar wind's $y'$-component distribution more positive.

We present in Fig. 6 the measurement distributions of $v_x$, $v_{y'}$, and $v_z$ in the $+E$ and $-E$ hemispheres as well as for all data. Since regular MHD flow interactions also deflect solar-wind protons in the magnetosheath, one strategy to determine if any additional deflection occurs due to $\mathbf{E} \times \mathbf{B}$ drift effects is to characterize the distributions with respect to a parameter related to this drift. The gyromotion equations in Behar et al. (2018) indicate that density ratios or velocity shears between protons and $O^+$ ions are the most appropriate parameters to investigate, however we cannot calculate these readily. Instead, we characterize

the distributions as a function of the magnitude the $y'$-component of the $\mathbf{E} \times \mathbf{B}$ velocity

$$|\mathbf{V}_{\mathbf{E} \times \mathbf{B}, y'}| = |\mathbf{V}_{\mathbf{E} \times \mathbf{B}, y}| = \left| \frac{(\mathbf{E} \times \mathbf{B})_y}{|\mathbf{B}|^2} \right| = \frac{|((-\mathbf{v}_{SW} \times \mathbf{B}) \times \mathbf{B})_y|}{|\mathbf{B}|^2} = \frac{|\mathbf{v}_{SW}||B_y \cos \theta_c|}{|\mathbf{B}|} = \frac{|\mathbf{v}_{SW}||\sin 2\theta_c|}{2}. \tag{2}$$



While not explicitly contained in the gyromotion equations, if the O$^+$ ions have a higher $\mathbf{V}_{\mathbf{E}\times\mathbf{B},y'}$ to which they can accelerate, then they will deflect the solar-wind protons more. We thus also separate the data into sets with high (above upper quartile), low (below lower quartile), and all values of $V_{\mathbf{E}\times\mathbf{B},y'}$. The $v_{y'}$ distributions in Fig. 6beh indeed shift towards more positive

values as $|\mathbf{V}_{\mathbf{E}\times\mathbf{B},y'}|$ increases, suggesting that the $\mathbf{E}\times\mathbf{B}$-related deflection observed at comet 67P is also measurable at Venus. The $v_z$ distributions exhibit no dependence on $|\mathbf{V}_{\mathbf{E}\times\mathbf{B},y'}|$, which makes sense since $\mathbf{V}_{\mathbf{E}\times\mathbf{B}}$ has no $z$ component. In contrast, $v_x$ increases in magnitude as $|\mathbf{V}_{\mathbf{E}\times\mathbf{B},y'}|$ increases, possibly reflecting less deceleration due to momentum exchange in the $x$ direction. However, the deceleration at the bow shock locally depends on $\theta_c$, so bow shock geometry effects should also be considered to explain this trend.

We note that the $v_x$ and $v_{y'}$ distributions appear more sensitive to $|\mathbf{V}_{\mathbf{E}\times\mathbf{B},y'}|$ in the $-E$ hemisphere. Given the higher prevalence of pick-up in the $+E$ hemisphere (Phillips et al., 1987; Barabash et al., 2007b; Jarvinen et al., 2013), we would expect a stronger coupling between these ions and the solar-wind protons in this hemisphere instead. Although Fig. 6 displays unnormalized data and the $-E$ hemisphere has higher plasma velocities, the distributions of the velocity components normalized by $|\mathbf{v}_{SW}|$ exhibit the same general features. The same is true when we characterize the distributions (normalized or not) as a

function of only $|\sin 2\theta_c|$. This shows that indeed the direction of planetary ion trajectories, which determines the direction of momentum exchange, affects the deflection of the solar-wind protons in a similar manner to what is observed at comet 67P. We wish to again quantify the asymmetry between hemispheres, however, the 'body' of the distribution of $v_{y'}$ in the $-E$ hemisphere shifts from mostly negative to mostly positive. The adequacy of the sum, difference, or ratio of the distributions as a measure of the asymmetry therefore becomes questionable. In any case, these preliminary results using the proxy parameter

$V_{\mathbf{E}\times\mathbf{B},y'}$ encourage further comparisons of $\mathbf{E}\times\mathbf{B}$ dynamics between Venus and comets, especially by including the O$^+$ bulk properties. Comparing to the simulations by Jarvinen et al. (2016) would also add to the discussion. Unfortunately, due to the Parker spiral angles and constant solar-wind speed used, the four cases all have nearly identical values of $|\mathbf{V}_{\mathbf{E}\times\mathbf{B},y'}|$, so we cannot incorporate them into this analysis.

## 5    Conclusions

Using measurements taken by Venus Express' ion mass-energy spectrometer and magnetometer, in this paper we characterized proton bulk-parameter asymmetries between the $\pm E$ hemispheres of Venus' dayside magnetosheath. The main results are as follows:

1. Both density and speed are slightly higher in the $-E$ hemisphere, whereas the magnetic-field strength is slightly higher in the $+E$ hemisphere. No significant asymmetries exist in the temperatures or their ratio.

2. The $y$ and $z$ components of the bulk velocity and their asymmetries exhibit trends with the upstream O$^+$ Larmor radius. Comparison to simulations and studies at Mars suggest these trends are consistent with deflection by momentum exchange with planetary ions.



**Figure 6.** Distributions of the proton bulk velocity components in the $+E$ (top row), $-E$ (middle row), and both (bottom row) hemispheres. Black is for data with high $|V_{\mathbf{E}\times\mathbf{B},y'}|$, dark gray for all $|V_{\mathbf{E}\times\mathbf{B},y'}|$, and light gray for low $|V_{\mathbf{E}\times\mathbf{B},y'}|$. The markers indicate the median for each distribution along with the respective first and third quartiles as the 'error' bars. The arrow indicates increasing $|V_{\mathbf{E}\times\mathbf{B},y'}|$.

3. The solar-wind deflection in the $y$ direction also exhibits trends with a proxy parameter related to the $\mathbf{E} \times \mathbf{B}$ drift of planetary ions. This again seems to result from momentum exchange with planetary ions and mimics solar-wind deflection dynamics observed at comet 67P.

Our comparative analysis between Venus, Mars, and comet 67P certainly has limitations yet nevertheless demonstrates the appeal of directly characterizing space plasmas across bodies of different scales. New studies jointly analyzing the plasma environment of the these bodies through simulation or observation with a uniform methodology would provide further insight into the phenomena discussed here. For example, identifying the parameters controlling the various asymmetries and characterizing their effect under equivalent upstream conditions can provide a more fundamental perspective of the solar-wind interaction with unmagnetized obstacles.

*Data availability.* All VEX data are publicly accessible at the ESA Planetary Science Archive at https://www.cosmos.esa.int/web/psa/venus-express. The dayside magnetosheath data are available in the in-text data citation Rojas Mata and Futaana (2022).

*Author contributions.* SRM analyzed the data, prepared the figures, and wrote the text. GSW and YF helped with text revision. TLZ is PI of VEX MAG.

*Competing interests.* The authors declare that they have no conflict of interest.

*Acknowledgements.* SRM was funded by the Swedish National Space Agency under contracts 145/19 and 79/19.



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
