# Peer review of "Proton Plasma Asymmetries between the Convective-Electric-Field Hemispheres of Venus' Dayside Magnetosheath"

_EGUsphere, 2023_

## Referee Comment (RC2)

**Review report on "Proton Plasma Asymmetries between the Convective-Electric-Field Hemispheres of Venus' Dayside Magnetosheath" by Sebastián Rojas Mata et al.**

In this work, the authors investigate plasma asymmetries in the Venusian magnetosheath. Using data from the Venus Express mission, they calculate mean values for proton density, velocity, temperature and magnetic field magnitude normalized to their solar wind counterparts. The authors take in hand a novel technique in order to calculate the plasma parameters, which later are organized according to the electric convective field. The main results do not show important asymmetries, but a secondary asymmetry on the velocity components leads to the second part of the paper where only the possible velocity asymmetries are studied. The final results are then compared with Mars and the comet 67P.

**Major comments:**
This reviewer finds a thoughtful analysis of the calculation of the moments, the choice of coordinate system, and the way the asymmetries are calculated for the observations. The authors claim that the clear asymmetries they found are for the proton density and the magnetic field strength with respect to the +E and -E hemispheres. This is not entirely true as the density shows very little asymmetry, while the magnetic field strength does not at all. So Conclusion 1 needs to be adjusted.

What follows from there, is an effort to look for an asymmetry showing an important dependence on the convective electric field which is translated in a possible relationship between the Larmor radius and the velocity components. However, the comparison is forced after imposing several constrictions. The authors do mention that comparing with the Mars case is not possible and they move on to compare with simulations. These last show a partial match with the observational results from the manuscript. Then comes the comparison between Venus and the comet 67P is a major concern. As a direct comparison is not possible, the authors impose one assumption after another to their own results to be able to compare with the 67P. The physical meaning of such a comparison would lack veracity. In this sense, Conclusions 2 and 3 are dubious.

The comparison with other non-magnetized bodies is not straightforward, even the authors are aware of it as expressed in the manuscript. I would suggest leaving out this section to a future work when a more direct comparison without too many assumptions could be done by the authors, which is -I believe- not the main point of this work and requires more work.

I have a few questions/comments on the first part of the manuscript, which is valuable material and I would recommend for publication after minor revisions.

**Other comments/questions:**
**1**. The data spatial coverage is limited for high latitudes and yet the authors calculated the asymmetries for the range 0-90°. Wouldn't it be better to leave out the regions with poor coverage using some criteria from the beginning? Perhaps something related to the time the spacecraft was in that region or a threshold in the number of measurements required to consider those angles.

**2**. Limited IMA field of view issue: did the authors take into account only full 3D distributions to calculate the plasma moments or did they also consider distributions with partial angular coverage? In the second case, how do the authors deal with the lack of physical meaning for partial distributions?

**3**. (Line 65) In the documentation for the IMA-MAG database it reads "*All solar wind (SW) parameters are medians calculated from IMA scans and MAG data not included in the file. These medians are assigned as the upstream SW conditions for each orbit (hence entries repeat for the same orbit).*"
The SW conditions change during each orbit, and the inbound leg of the orbit is unlikely to be preceded by the same SW conditions as the outbound leg. If one considers two data points, one closer to the shock in the inbound leg and the other closer to the shock in the outbound leg, it does not make sense to consider the same set of upstream SW conditions for both points (either upstream inbound or upstream outbound). One point should be associated with the closest SW region mapped by the spacecraft. So could you clarify which SW conditions are considered for an orbit: upstream inbound leg or upstream outbound leg? How do you choose which to use? If the exact same upstream SW conditions are taken no matter where the measurement point is, what is the effect of such a selection on the calculated asymmetries?

**4**. (Line 109) "*The perpendicular and parallel temperatures exhibit the clearest symmetry between hemispheres (both overall and as a function of latitude).*"
In Figure 1, the asymmetries for Tpar and Tperp are clear; however, the parameter asymmetry in Figure 2 is visible for Tperp, but that is not true for Tparallel. Why is it so?

**5**. (Line 134) If an artifact effect, shouldn't it be also shown in the third quartile and for all data?

**6**. (Line 195) So the asymmetry for large Larmor radius is related to the fact that ions are picked-up more easily in that direction than in others? But, I would expect for this to happen far from the planet as $r_L$ is large, therefore the asymmetry will be very hard to observe close to the planet.

---

## Author Comment (AC1)

**Response Referee Comment 1 (RC1): Paper EGUSPHERE-2023-2570**

**Sebastián Rojas Mata**

**April 12, 2024**

Dear Anonymous Referee #1,

Thank you for your time reading and commenting our manuscript "Proton Plasma Asymmetries between the Convective-Electric-Field Hemispheres of Venus' Dayside Magnetosheath" for publication in *Annales Geophysicae*. Your feedback provided valuable avenues to revise and improve our manuscript. Below we list your comments in italic text, followed by our responses in normal text.

*The paper is highly technical and not the easiest to read, probably requiring that the reader read several of the preceding work. This isn't a red card, this is how science works, but I want to flag that it makes this particular paper quite hard going. This review is written from the point of view of someone familiar with Venus Express and IMA, but not with the previous research in this specific field. Apologies for any misunderstandings, this paper is quite dense.*

Based on this and other feedback, we will shorten/exclude parts of Section 4, which will provide space to expand Section 2 and elaborate on the technical background without making the paper longer.
* * *
**Q 1.1** *The introduction of paper would benefit greatly from a cartoon or simple explanatory figure describing the overall geometry of the Venusian magnetosphere, the different hemispheres, the actual regions being studied. While I am sure this is second nature to the authors, the paper requires a developed 3D mental picture of the induced magnetosphere of Venus and I am concerned will be quite difficult to follow for any reader who does not already have this in their mind.*

**Reply**: A diagram illustrating the near-Venus environment (regions, boundaries, etc.) and the electromagnetic fields defining the VSE reference frame will be added in the introduction.
* * *
**Q 1.2** *This is a highly technical paper. The brief description of the novel methodology used to calculate the data products used is quite terse and the paper would stand alone a lot better if it could hold the readers hand through the process. All I got from it was that some form of gaussian distribution is being assumed and some sort of curve fitting is going on? It is not very clear. If so, how have the authors ensured that the field of view obscurations of IMA have not skewed results?*

**Reply**: We will expand our summary of the methodologies developed in previous papers. Regarding FOV blockage, the use of Gaussian fits instead of moment integrals helps 'fill out' the proton VDF whenever its sampling was not perfect (to an extent). This study uses a subset of all IMA measurements which were filtered through a variety of quality metrics to better ensure that poor measurements are excluded from the statistical analysis. The previous papers address this and similar concerns so we will make sure to mention them in our summary of methodology.
* * *
**Q 1.3** *Regarding Fig 1, I appreciate that the authors have clarified that the 'error' bars are actually first and third quartiles with the central square presumably as the median of the distribution. From this, as a first time reader, what I take away is that all the data are mostly the same for all parameters across all VSE latitudes. No asymmetries appear present in the data. The authors claim an asymmetry in velocity, and I do indeed see how it seems to trend to higher velocities at high negative latitudes.*

*"The plasma speed appears lower closer to the central parallel". What is the central parallel? If this is*

*VSE 0o, can you please throw the reader a lifeline and say this? Am I supposed to be looking at both the top and bottom panels, or am I just supposed to be looking at the bottom panels (normalized) with the top as reference? If so, why?*

*If this is what is being claimed, I have a few concerns with the claim of a trend.*

*All data points appear to overlap within 'errors'. How can a trend be statistically claimed? I have exactly the same concern for Fig 6 of Rojas Mata et al. (2023) which this paper relies upon.*

*In the raw data (panel b) the apparent trend comes mostly from the bottom few data points, beginning at but, as the authors themselves say, "Bins 75° or farther contain much fewer scans (< 25) so these data may have lower statistical reliability." If I cover these up, then the 'trend' is far less clear to my eye.*

*In the normalized data (panel g), again, covering up the bottom 3 data points where the statistics are dubious, I am also unconvinced that a trend exists. Please perform a rigorous and appropriate statistical test on the binned data to prove that the trends are real and not just a random pattern from poorer sampling that has been biased by some unknown reason. If the authors have doubt about the statistical veracity of data at 75 degrees and greater, then they could consider cutting off their analysis at some threshold latitude that was better supported by the data and where the statistical sampling is significant enough.*

*The claim "This indicates that the convective electric field has little influence on average magnetosheath properties, especially compared to the bow shock geometry." Is similarly very strong. An alternative conclusion could be that there's simply too much noise in the data to extract any meaningful trends. I recommend further statistical analysis to demonstrate statistically that the distributions are the same, and give the certainty to this.*

*The conclusion that "Both density and speed are slightly higher in the -E hemisphere, whereas the magnetic-field strength is slightly higher in the +E hemisphere" does not appear to be supported by the data. Density appears constant in both hemispheres (Fig 1a,g). Likewise with Magnetic field strength (Fig c,i, especially if one ignores the data points at high latitudes which the authors have flagged as having poor statistics). I thus cannot agree with this conclusion at present.*

**Reply**: We agree that trends with latitude are weak (or non-existent) and the sentence in lines 98-99 is meant to reflect this; we will clarify this point. However, the statements concerning asymmetries are based on Figure 2, not Figure 1, since in line 99 we move on from Figure 1 to discuss the results of the computed asymmetries. The comments about what the data looks like in Figure 1acgi are not applicable to conclusions about asymmetries.

These comments however have provided good opportunity to review how we present our results in the context of the previous asymmetry studies. We will thus reorder the paragraph in lines 97-111 to state more directly that our analysis does not reveal as conclusive asymmetries as previous work has. The reviewer has understandable concerns that this could be due to noise or poor statistics, however the fact that the same data set analyzed with the same methodology revealed clear asymmetries between the quasi-perpendicular and quasi-parallel magnetosheaths (Figures 7 and 8 in Rojas Mata et al. 2023) suggests otherwise. We mention in lines 115-116 that we sought trends in the two hemispheres with respect to upstream O+ Larmor radius and found nothing significant. We also tried this with other parameters (density, speed, Alfvén Mach number, etc.) yet none of the scalar parameters showed significant trends. If characterizing the plasma by which E hemisphere it is in does not reveal significant trends, then this is not a significant controlling parameter (at least by itself) of the magnetosheath plasma state. We have also excluded data in the three highest-latitude bins on both sides when calculating overall asymmetries and found that it has no discernible effect on the result. We will mention this in the manuscript as it is related to the suggestion about cutting off data at a certain latitude.
* * *
*Q 1.4 The conclusion that "The y and z components of the bulk velocity and their asymmetries exhibit trends with the upstream O+ Larmor radius." Is in apparent contradiction to the earlier acknowledgment that "IMA's limited FOV combined with the spacecraft's orientation sometimes leads to measurements with*

*poor constraints on vz. Reviewing the measurements to correct vz is beyond the scope of this work". If the data is suspect, can the authors please explain why conclusions are being drawn upon it?*

**Reply**: Our wording in the manuscript potentially comes off too harsh and arises more concern than it should. It is true that usually the vz measurements are less well constrained compared to the vx and vy measurements, but not to the point of serious concern. The majority of vz values have the sign we would expect them to have in each hemisphere as reflected by the quartiles marked in Figure 3. This expectation may also be too restrictive and having some positive (negative) vz measurements in the -E (+E) hemisphere is not unrealistic. Romanelli et al. (2020) only discuss the means of vz and no other study we have found includes measurements of vz, so unfortunately we cannot compare to related work for a sanity check on these measurement distributions (this is why in lines 180-183 we advocate for an analogous study at Mars).

We could exclude or flip the sign of the vz measurements with the 'wrong' sign, however this seems like an unjustifiable (or maybe even dishonest) way of manipulating the data. We did check that doing either of these things yields qualitatively similar trends with O+ Larmor radius, however we prefer to report and work with the measurement distributions shown in Figure 3. We will rephrase our critique of the impact of IMA's FOV and spacecraft orientation so that it does not generate unnecessary concern in the reader.
* * *
***Q 1.5*** *The caption of Fig 1 should take the opportunity to clarify once again that these are magnetosheath measurements.*

**Reply**: We will include this reminder.
* * *
***Q 1.6*** *How have the authors ensured that data were selected only from the magnetosheath?*

**Reply**: The IMA scans were labeled as being in the solar wind or the magnetosheath by manually determining the location of dayside bow-shock crossing for each orbit. We will make this clear when expanding on methodology as mentioned in Q 1.2.

---

## Author Comment (AC2)

**Response Referee Comment 2 (RC2): Paper EGUSPHERE-2023-2570**

Sebastián Rojas Mata

April 12, 2024

Dear Anonymous Referee #2,

Thank you for your time reading and commenting our manuscript "Proton Plasma Asymmetries between the Convective-Electric-Field Hemispheres of Venus' Dayside Magnetosheath" for publication in *Annales Geophysicae*. Your feedback provided valuable avenues to revise and improve our manuscript. Below we list your comments in italic text, followed by our responses in normal text.

*In this work, the authors investigate plasma asymmetries in the Venusian magnetosheath. Using data from the Venus Express mission, they calculate mean values for proton density, velocity, temperature and magnetic field magnitude normalized to their solar wind counterparts. The authors take in hand a novel technique in order to calculate the plasma parameters, which later are organized according to the electric convective field. The main results do not show important asymmetries, but a secondary asymmetry on the velocity components leads to the second part of the paper where only the possible velocity asymmetries are studied. The final results are then compared with Mars and the comet 67P.*
* * *
**Q 1.1** *This reviewer finds a thoughtful analysis of the calculation of the moments, the choice of coordinate system, and the way the asymmetries are calculated for the observations. The authors claim that the clear asymmetries they found are for the proton density and the magnetic field strength with respect to the +E and -E hemispheres. This is not entirely true as the density shows very little asymmetry, while the magnetic field strength does not at all. So Conclusion 1 needs to be adjusted.*

**Reply**: We will revise Conclusion 1 to be quantitative (instead of just saying 'slight') and contrast with what previous studies observed.
* * *
**Q 1.2** *What follows from there, is an effort to look for an asymmetry showing an important dependence on the convective electric field which is translated in a possible relationship between the Larmor radius and the velocity components. However, the comparison is forced after imposing several constrictions. The authors do mention that comparing with the Mars case is not possible and they move on to compare with simulations. These last show a partial match with the observational results from the manuscript. Then comes the comparison between Venus and the comet 67P is a major concern. As a direct comparison is not possible, the authors impose one assumption after another to their own results to be able to compare with the 67P. The physical meaning of such a comparison would lack veracity. In this sense, Conclusions 2 and 3 are dubious.*

*The comparison with other non-magnetized bodies is not straightforward, even the authors are aware of it as expressed in the manuscript. I would suggest leaving out this section to a future work when a more direct comparison without too many assumptions could be done by the authors, which is -I believe- not the main point of this work and requires more work.*

**Reply**: We greatly appreciate the feedback on the comparison with other non-magnetized bodies since this is the riskiest part of our analysis. It really is unfortunate that analogous studies at Mars or comets don't exist since they would provide valuable material for a great comparative-planetology discussion. Given the feedback, we plan to remove Section 4.3 entirely and rework Sections 4.1 and 4.2 into a shorter discussion which invites for further investigation of the topic instead of drawing such strong conclusions. Thus Conclusion 3 will be removed and the second part of Conclusion 2 will be revised. Figures 3 and 4 will be kept

since they provide a useful exploratory analysis of Larmor-radius dependencies that a future study can delve into. Cutting out Section 4.3 will also open up space for expanding on methodology in Section 2 without making the paper longer.
* * *
**Q 1.3** *The data spatial coverage is limited for high latitudes and yet the authors calculated the asymmetries for the range 0-90°. Wouldn't it be better to leave out the regions with poor coverage using some criteria from the beginning? Perhaps something related to the time the spacecraft was in that region or a threshold in the number of measurements required to consider those angles.*

**Reply**: We calculated the overall asymmetries excluding measurements at high latitudes (75° or more from central parallel) and the results don't change since there are much fewer scans there. Applying some criteria to exclude low-coverage regions is a good suggestion but not necessary for the analysis. We will mention in the manuscript that data from the low-coverage regions does not affect the final results or conclusions.
* * *
**Q 1.4** *Limited IMA field of view issue: did the authors take into account only full 3D distributions to calculate the plasma moments or did they also consider distributions with partial angular coverage? In the second case, how do the authors deal with the lack of physical meaning for partial distributions?*

**Reply**: The plasma parameters were not calculated through moment integrals but rather through the VDF fitting procedure developed by Bader et al. (2019). This has the advantage of 'filling out' the VDF measurement whenever IMA did not measure the full distribution. This is only reasonable to an extent so the procedure includes quality metrics to filter out poor fits. We will include more details like this when we expand on methodology in Section 2.
* * *
**Q 1.5** *(Line 65) In the documentation for the IMA-MAG database it reads "All solar wind (SW) parameters are medians calculated from IMA scans and MAG data not included in the file. These medians are assigned as the upstream SW conditions for each orbit (hence entries repeat for the same orbit)."*
*The SW conditions change during each orbit, and the inbound leg of the orbit is unlikely to be preceded by the same SW conditions as the outbound leg. If one considers two data points, one closer to the shock in the inbound leg and the other closer to the shock in the outbound leg, it does not make sense to consider the same set of upstream SW conditions for both points (either upstream inbound or upstream outbound). One point should be associated with the closest SW region mapped by the spacecraft. So could you clarify which SW conditions are considered for an orbit: upstream inbound leg or upstream outbound leg? How do you choose which to use? If the exact same upstream SW conditions are taken no matter where the measurement point is, what is the effect of such a selection on the calculated asymmetries?*

**Reply**: The magnetosheath measurements we consider are all located in the dayside. We thus use the solar-wind measurements upstream of the dayside bow-shock crossing for each orbit. These can be during the inbound or outbound leg, but always immediately before or after the magnetosheath scans. We will make this clearer when we expand on methodology in Section 2.
* * *
**Q 1.6** *(Line 109) "The perpendicular and parallel temperatures exhibit the clearest symmetry between hemispheres (both overall and as a function of latitude)."*
*In Figure 1, the asymmetries for Tpar and Tperp are clear; however, the parameter asymmetry in Figure 2 is visible for Tperp, but that is not true for Tparallel. Why is it so?*

**Reply**: Judging asymmetry from Figure 1 is not appropriate since it corresponds to taking a ratio of medians instead of a median of ratios. The asymmetries shown in Figure 2 are the statistically superior way to calculate ratios since they are based on the entire distributions, not just a couple representative values like the quartiles. This advantage of the method is mentioned in Rojas Mata et al. (2023) and we will mention it again when we expand on methodology in Section 2.
* * *
**Q 1.7** *(Line 134) If an artifact effect, shouldn't it be also shown in the third quartile and for all data.*

**Reply**: We are not exactly sure what is meant here. The third quartile for small Larmor-radii vx in the -E hemisphere is more to the left than the other two third quartiles which is what the small dip at -100

km/s would cause. The data for all Larmor radii do not show this dip likely because the other 75% of measurements don't feature it.
* * *
**Q 1.8** *(Line 195) So the asymmetry for large Larmor radius is related to the fact that ions are picked-up more easily in that direction than in others? But, I would expect for this to happen far from the planet as $r_L$ is large, therefore the asymmetry will be very hard to observe close to the planet.*

**Reply**: Simulations like those in Jarvinen et al. (2016) would provide a great opportunity to characterize the spatial extent of the asymmetry by calculating analogous latitudinal dependencies and maybe accompanying radial dependencies. Our results suggest that measurable differences in the solar-wind deflection occur close to the planet. Maybe this is due to the pick-up ions having much greater mass than the solar wind so even a little bit of transferred momentum noticeably deflects the solar wind. On the other hand, the relative densities also need to be considered since these also determine the total momentum of each species flow. A dedicated study analyzing data and simulations in the same way would provide important insight to these Larmor-radius trends.